# Learning Disentangled Multi-Agent World Model for Decentralized Control

## Abstract

World models enable learning policies via latent imagination, offering benefits such as history compression and sample efficiency. The primary challenge in applying world models to multi-agent tasks is that modeling multi-agent dynamics in latent space requires integrating information from different agents, often creating spurious correlations between their latent states. Existing methods either reconstruct the observation for each agent or employ communication to maintain correlation during execution, failing to learn disentangled latent states that are crucial for effective decentralized control. To address this, we present the Disentangled Multi-Agent World Model (DMAWM). It facilitates learning decentralized policies in the latent space through a novel architecture comprising independent agent modules and a shared environment module. During real-environment execution, agent modules independently process local information to form a factorized latent representation. The environment module is then trained to mirror the factorized structure generated by the agent modules, effectively disentangling individual latent states from the interaction dynamics. Consequently, imaginary rollouts generated by the environment module more faithfully simulate decentralized execution dynamics, facilitating the transfer of policies from imagination to decentralized execution. Empirically, DMAWM outperforms existing model-based and model-free approaches in convergence speed and final performance, with additional visualization demonstrating its efficacy in capturing agent interactions.

## 1 Introduction

Model-based reinforcement learning (MBRL) has emerged as a highly effective approach in the single-agent domain. It has enabled superhuman performance in complex tasks such as Atari and board games without prior knowledge of the rules (Schrittwieser et al., 2020), and has allowed agents to master challenging objectives like collecting diamonds in Minecraft from scratch (Hafner et al., 2025). The cornerstone of these successes lies in learning models of the environments, often referred to as world models (Ha & Schmidhuber, 2018). The world model summarizes the agent's history of observations and actions into a compact latent state representation. This representation is predictive of future outcomes, like subsequent observations or rewards, enabling agents to efficiently plan or learn policies through simulated experience (imagination) within the learned world model.

However, extending world models to multi-agent reinforcement learning (MARL) is challenging. A primary difficulty lies in that modeling multi-agent dynamics in latent space requires integrating information from different agents, leading to spurious correlations that are impossible to maintain during decentralized execution. This hinders the learning of disentangled latent states crucial for effective decentralized control. Existing model-based MARL methods have attempted to address these challenges with varying degrees of success. Some generate trajectories directly in the original observation space (Zhang et al., 2021; Xu et al., 2022; Zhang et al., 2024), often struggling with high-dimensional observations and neglecting the benefits of history-dependent latent states. Others employ communication to maintain the correlation during execution (Egorov & Shpilman, 2022; Wu et al., 2023; Toledo, 2024). However, they often compromise the requirement for fully decentralized execution and fail to learn the disentangled latent states crucial for decentralized policies.

To address this, we introduce Disentangled Multi-Agent World Model (DMAWM), a novel framework to learn decentralized policies effectively within a latent space. It has a distinctive architecture

with independent agent modules and a shared environment module. During real-world execution, the agent modules independently process local observations to form a factorized latent representation across all agents. Crucially, the environment module is then trained to mirror the factorized structure generated by the agent modules, which effectively disentangles the individual latent states of each agent. This disentanglement ensures that the imaginary rollouts generated by the environment module faithfully simulate decentralized execution dynamics. Consequently, policies learned by imagination within the model can be transferred to the real environment in a decentralized manner.

We evaluate DMAWM on three challenging MARL benchmarks, including SMAC (Samvelyan et al., 2019), SMACv2 (Ellis et al., 2023), and vision-based Melting Pot (Egorov & Shpilman, 2022). DMAWM outperforms existing model-based and model-free MARL baselines in convergence speed and final performance, demonstrating its effectiveness in learning multi-agent dynamics to facilitate effective decentralized policy learning. We also conduct visualization analysis to demonstrate DMAWM's efficacy in capturing agent interactions in the latent space.

## 2 RELATED WORK

**World models for control** World models learn the underlying dynamics from raw observations for data-efficient control. Generative world models are optimized to reconstruct the observation to capture the dynamics: SimPLe (Kaiser et al., 2020) trains a video predictor as an environment model, and the Dreamer family (Hafner et al., 2020; 2021; 2025) leverages recurrent state-space models (RSSMs) (Hafner et al., 2019) to plan in latent space. Implicit world models avoid decoding observations and instead couple learned dynamics with decision procedures—MuZero (Schrittwieser et al., 2020) and EfficientZero (Ye et al., 2021) integrate MCTS, while TD-MPC and TD-MPC2 (Hansen et al., 2022; 2024) pair implicit models with MPC for continuous control. Recent advances focus on stronger backbone models, bringing transformers and diffusion into dynamics learning: IRIS (Micheli et al., 2023) and TWM (Robine et al., 2023) apply transformers to model the whole trajectory in an end-to-end manner, while UniSim (Yang et al., 2024) and DIAMOND (Alonso et al., 2024) employs diffusion models to enhance the capability of capturing the visual details.

Our work is built on the latent generative world model due to its efficiency, but targets multi-agent settings by explicitly factorizing dynamics into agent and environment modules, enabling structured imagination of interactions and supporting decentralized execution.

**Model-based MARL** Model-based MARL must handle partial observability of the environment and non-stationarity from concurrently adapting policies. Early approaches assume global observability (Krupnik et al., 2020; Zhang et al., 2021) or sufficiency of joint observations (Willemsen et al., 2021), sidestepping history dependencies. Later works pose learning multi-agent dynamics as sequence modeling task using recurrent networks (Xu et al., 2022) or transformers (Zhang et al., 2024; Liu et al., 2024), facilitating long-horizon imagination but often struggling with high-dimensional observations. Inspired by Dreamer, latent world models have been adapted to MARL; however, many methods rely on broadcasting the latent states to other agents to keep latent states consistent across imagination and execution: MAMBA (Egorov & Shpilman, 2022) maintains local world models with communication; MAG (Wu et al., 2023) further mitigates multi-step errors in world modeling; CoDreamer (Toledo, 2024) utilizes GNN-based communication for both latent state update and action selection. MABL (Venugopal et al., 2024) proposes a bi-level architecture to reduce communication requirements, but its global state transition model may not scale effectively as the number of agents increases.

Unlike previous methods that reconstruct observations or broadcast latent states, our work forces the latent states during imagination to reflect the factorized structure as in the real environment. This design leads to fully decentralized execution while enabling the world model to model the interactions between agents without the need for learning a global state transition model.

## 3 PRELIMINARY

**Learning latent dynamics** Recurrent state-space models (RSSMs) (Hafner et al., 2019; 2020) learn action-conditioned generative dynamics that can roll out trajectories of observations and rewards given actions. An RSSM (parameterized by $\phi$) comprises: (1) a representation model

$q_\phi(I_t \mid I_{t-1}, a_{t-1}, o_t)$[1] that infers the posterior latent state from the previous latent state $I_{t-1}$, the previous action $a_{t-1}$, and the current observation $o_t$; (2) a transition model $p_\phi(\hat{I}_t \mid I_{t-1}, a_{t-1})$ that predicts the next latent state from the previous state and action; and (3) decoders $p_\phi(\hat{o}_t \mid I_t)$ and $p_\phi(\hat{r}_t \mid I_t)$ that reconstruct the observation and reward from the latent state. These components are jointly trained to maximize the evidence lower bound (ELBO) on observed trajectories. This objective encourages high likelihood for the observed trajectories, while enforcing the consistency between the representation model and the transition model to capture the environment dynamics. During policy training, imagined trajectories are generated by first initializing $I_1$ from a subsequence of an observed trajectory sampled from the replay buffer, and then recursively predicting latent states $\hat{I}_{t+1} \sim p_\phi(\cdot \mid I_t, a_t)$ with action sampled from a learned policy $a_t \sim \pi_\theta(\cdot \mid I_t)$.

DreamerV2 (Hafner et al., 2021) further decomposes the latent state $I_t = (h_t, z_t)$ into a deterministic component $h_t$ and a stochastic component $z_t$. Given an observation $o_t$, the agent first updates the deterministic state via the recurrent model $h_t = f_\phi(I_{t-1}, a_{t-1})$, and then infers the stochastic state with the representation model $z_t \sim q_\phi(\cdot \mid h_t, o_t)$. During imagination, it reuses the recurrent model to update the deterministic state $h_t = f_\phi(I_{t-1}, a_{t-1})$, but samples the stochastic component from a separate model $\hat{z}_t \sim p_\phi(\cdot \mid h_t)$ without access to observations.

**Dec-POMDPs**  We consider cooperative, partially observable tasks formalized as decentralized partially observable Markov decision processes (Dec-POMDPs) (Oliehoek & Amato, 2016). In a Dec-POMDP with $n$ agents, each agent $i$ acts based on local information. At each discrete timestep $t$, while the environment is in state $s_t$, agent $i$ receives a local observation $o_t^i \sim p(\cdot \mid s_t)$, which is appended to its local history $\tau_t^i = (o_{1:t}^i, a_{1:t-1}^i)$. Based on this history, the agent selects an action $a_t^i \sim \pi^i(\cdot \mid \tau_t^i)$, forming a joint action $a_t^{1:n} = (a_t^1, \ldots, a_t^n)$. Executing $a_t^{1:n}$ in state $s_t$ transitions the environment to $s_{t+1}$ and yields a shared reward $r_{t+1}$ according to $p(s_{t+1}, r_{t+1} \mid s_t, a_t^{1:n})$. Without loss of generality, we can extend this shared reward to agent-specific rewards $r_t^{1:n} = (r_t^1, \ldots, r_t^n)$, where for cooperative tasks, all agents share the same reward: $r_t^1 = \ldots = r_t^n = r_t$. The goal is to learn decentralized policies $\pi^{1:n} = (\pi^1, \ldots, \pi^n)$ that maximize the expected discounted return $\mathbb{E}_{\pi^{1:n}, p}[\sum_{t=1}^{\infty} \gamma^{t-1} r_{t+1}]$, where $\gamma \in (0, 1)$ is the discount factor.

# 4 METHOD

We present DMAWM, a framework for learning decentralized policies in latent space. We first introduce its overall design, which features independent agent modules and a shared environment module (Section 4.1). Next, we describe how it enables effective learning of multi-agent dynamics while preserving the factorized structure (Section 4.2). Finally, we explain how the disentangled latent states facilitate training decentralized policies through imagined trajectories (Section 4.3).

## 4.1 FRAMEWORK

DMAWM features a distinctive architecture comprising independent agent modules and a shared environment module, as shown in Figure 1. This design enables effective learning of disentangled latent states while capturing agent interactions. The agent modules independently process local observations to form factorized latent representations, while the environment module models agent interactions during imagination, generating imaginary trajectories for policy learning.

**Agent module**  Each agent module (parameterized by $\psi$) operates independently, maintaining its own internal state $I_t^i = (h_t^i, z_t^i)$ that consists of a deterministic component $h_t^i$ and a stochastic component $z_t^i$. Upon receiving a local observation $o_t^i$, the agent updates its state through two components: the recurrent model updates the deterministic part $h_t^i = f_\psi(I_{t-1}^i, a_{t-1}^i)$, while the representation model infers the stochastic part $z_t^i \sim q_\psi(\cdot \mid h_t^i, o_t^i)$, forming a factorized posterior $q_\psi(z_t^{1:n} \mid h_t^{1:n}, o_t^{1:n}) = \prod_{i=1}^{n} q_\psi(z_t^i \mid h_t^i, o_t^i)$, as the individual stochastic state is only conditioned on local information. Action selection is based solely on the agent's internal state, i.e., $a_t^i \sim \pi_\theta(\cdot \mid I_t^i)$, ensuring decentralized decision-making.

---

[1]We use the notation $I_t$ to emphasize it represents the internal state of an agent, which corresponds to $s_t$ in the original RSSM paper (Hafner et al., 2019).

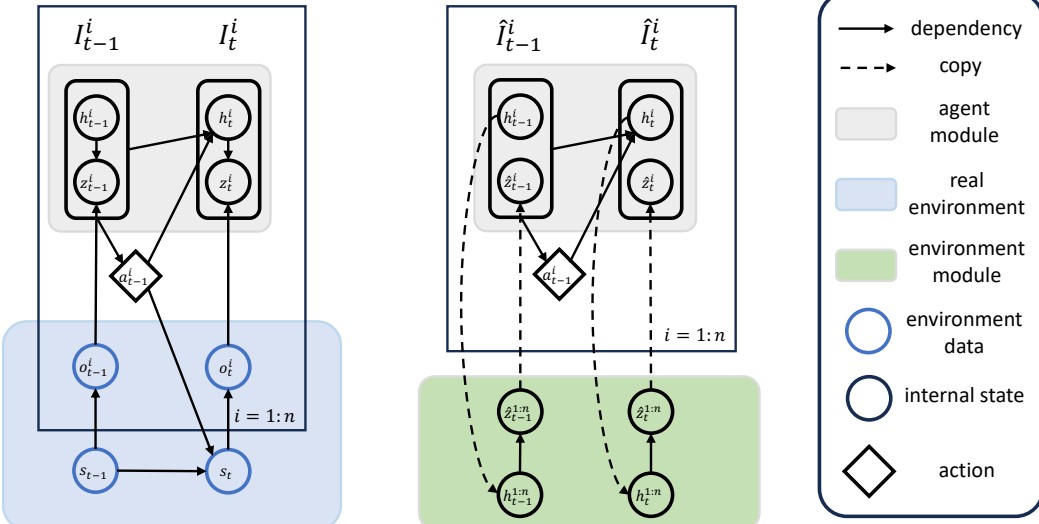

(a) Interaction with real environment     (b) Interaction with a shared environment module

Figure 1: Illustration of agent interactions. The gray area represents the agent module where the latent states are updated; the blue area represents the real environment which manages the state transition and observation generation; the green area represents the environment module which replaces the real environment during imagination and couples agents via their latent states.

**Environment module**   The environment module (parameterized by $\phi$) plays a crucial role in generating imaginary trajectories for policy learning. During imagination, it replaces the real environment by coupling the agents' internal states to simulate environment dynamics and inter-agent interactions. Given the previous joint latent state $I_{t-1}^{1:n}$ and actions $a_{t-1}^{1:n}$, the recurrent models of agent modules independently compute their deterministic components $h_t^i = f_\psi(I_{t-1}^i, a_{t-1}^i)$. The key innovation is a transformer-based interaction predictor which models inter-agent interactions by sampling the joint prior $p_\phi(\hat{z}_t^{1:n} \mid h_t^{1:n})$ for all agents. Intuitively, the interaction predictor should capture the necessary dependencies between agents to model the evolution of the latent states in decentralized execution. To achieve this, we align this joint prior to the factorized posterior $\prod_{i=1}^n q_\psi(z_t^i \mid h_t^i, o_t^i)$ via the dynamics loss and representation loss (introduced in Section 4.2). In addition, the environment module includes decoders that reconstruct trajectory components from the latent states, providing supervision signals for representation learning and policy training.

The agent module and the environment module consist of:

$$\text{Agent module} \begin{cases} \text{Recurrent model:} & h_t^i = f_\psi(h_{t-1}^i, z_{t-1}^i, a_{t-1}^i), \\ \text{Representation model:} & z_t^i \sim q_\psi(\cdot \mid h_t^i, o_t^i) \end{cases} \tag{1}$$

$$\text{Environment module} \begin{cases} \text{Interaction predictor:} & \hat{z}_t^{1:n} \sim p_\phi(\cdot \mid h_t^{1:n}) \\ \text{Observation decoder:} & \hat{o}_t^i \sim p_\phi(\cdot \mid h_t^i, z_t^i) \\ \text{Reward decoder:} & \hat{r}_t^{1:n} \sim p_\phi(\cdot \mid h_t^{1:n}) \\ \text{Continuation decoder:} & \hat{c}_t^{1:n} \sim p_\phi(\cdot \mid h_t^{1:n}) \\ \text{Available actions decoder:} & \hat{m}_t^{1:n} \sim p_\phi(\cdot \mid h_t^{1:n}) \end{cases} \tag{2}$$

where for each agent $i$, we use the binary vector $\hat{m}_t^i$ to denote the available actions, and the binary scalar $\hat{c}_t^i$ to denote the continuation flag with $\hat{c}_t^i = 1$ if the episode continues at $t$.

### 4.2   LEARNING DISENTANGLED MULTI-AGENT WORLD MODEL

The learning objective of our multi-agent latent dynamics model is designed to maintain the factorized structure while effectively capturing agent interactions. Similar to Dreamer (Hafner et al.,

2020), the training process iterates between collecting real environment data, learning the latent dynamics model, and training policies through imagination.

To train the multi-agent world model, we optimize the following objective:

$$\mathcal{L}(\phi, \psi) = \underbrace{\beta_{\text{dyn}} \mathcal{L}_{\text{dyn}}(\phi, \psi)}_{\text{dynamics loss}} + \underbrace{\beta_{\text{rep}} \mathcal{L}_{\text{rep}}(\psi)}_{\text{representation loss}} + \underbrace{\beta_{\text{dec}} \mathcal{L}_{\text{dec}}(\phi, \psi)}_{\text{decoder loss}}, \tag{3}$$

where $\beta_{\text{dyn}}$, $\beta_{\text{rep}}$, and $\beta_{\text{dec}}$ are the weights for each loss term. The dynamics loss and representation loss align the prior and posterior distribution of the latent states, while the decoder loss encourages the model to reconstruct the real trajectories. Together, the above objective maximizes the variational lower bound of the likelihood of the real trajectories. The detailed derivation is in Appendix A.2.

**Dynamics loss and representation loss**    The dynamics and representation losses enforce disentanglement of latent states while still capturing their interactions during imagination. By disentanglement, we mean that posterior latent states are conditionally independent given agents' observations, i.e., $q_\psi(z_t^{1:n} \mid h_t^{1:n}, o_t^{1:n}) = \prod_{i=1}^{n} q_\psi(z_t^i \mid h_t^i, o_t^i)$. Thus, the interaction between agents is modeled only through the observations. Since no observation is available during imagination, the interaction predictor takes the joint deterministic state as input to model the interaction between agents instead. By aligning the joint prior with the factorized posterior, we compel the interaction predictor to generate a statistically consistent joint state, even in the absence of observations. We train the prior and the posterior via the following losses:

$$\mathcal{L}_{\text{dyn}}(\phi, \psi) = \sum_{t=1}^{T} D_{\text{KL}} \left( \prod_{i=1}^{n} \text{sg}(q_\psi(\cdot \mid h_t^i, o_t^i)) \parallel p_\phi(\cdot \mid h_t^{1:n}) \right), \tag{4}$$

$$\mathcal{L}_{\text{rep}}(\psi) = \sum_{t=1}^{T} D_{\text{KL}} \left( \prod_{i=1}^{n} q_\psi(\cdot \mid h_t^i, o_t^i) \parallel \text{sg}(p_\phi(\cdot \mid h_t^{1:n})) \right), \tag{5}$$

where $\text{sg}(\cdot)$ denotes the stop-gradient operator and $T$ is the trajectory length. The dynamics loss $\mathcal{L}_{\text{dyn}}$ trains the interaction predictor to match the factorized posterior, while the representation loss $\mathcal{L}_{\text{rep}}$ regularizes the posterior toward the joint prior. Using different weights for the two losses allows us to use higher weight for optimizing the prior. $\mathcal{L}_{\text{dyn}}$ optimizes both $\phi$ and $\psi$ as $h_t^{1:n}$ is differentiable with respect to $\psi$.

**Decoder loss**    The decoders are trained to reconstruct the real trajectories, enabling the model to learn an informative representation of the environment and the structure of the underlying Dec-POMDP. The observation decoder reconstructs observations by taking both deterministic and stochastic states as input. Decoding the rewards, continuation flags, and available actions often require additional information beyond the local history, for example, an episode terminates when either all agents or all enemies are eliminated. To address this, we first combine all agents' deterministic states through a shared self-attention block, then decode the rewards, continuation flags, and available actions using separate heads. The decoder loss comprises:

$$\mathcal{L}_{\text{dec}}(\phi, \psi) = -\sum_{t=1}^{T} \left( \sum_{i=1}^{n} \log p_\phi(o_t^i \mid h_t^i, z_t^i) + \log p_\phi(r_t^{1:n} \mid h_t^{1:n}) \right.$$
$$\left. + \log p_\phi(c_t^{1:n} \mid h_t^{1:n}) + \log p_\phi(m_t^{1:n} \mid h_t^{1:n}) \right), \tag{6}$$

where $\{o_t^{1:n}, r_t^{1:n}, c_t^{1:n}, m_t^{1:n}\}_{t=1}^{T}$ are the ground-truth trajectory components. $\mathcal{L}_{\text{dec}}$ optimizes both $\phi$ and $\psi$ as $h_t^{1:n}$ is differentiable with respect to $\psi$. The loss in equation 3 is optimized end-to-end with backpropagation through time (BPTT), with the gradient of discrete latent states estimated by the straight-through estimator (Bengio et al., 2013).

**Absorbing state**    In multi-agent tasks, agents can become absent due to death or leaving the scene (Samvelyan et al., 2019; Li et al., 2022). Drawing inspiration from prior work (Schrittwieser et al., 2020; Egorov & Shpilman, 2022), we address this by incorporating an absorbing state into our latent dynamics model to represent agent absence. When an agent is absent, it immediately

transitions to the absorbing state and remains there indefinitely, where the agent constantly observes a fixed null observation and executes a fixed non-operation action. When an episode is terminated in the environment, we let all agents remain in the absorbing state in the latent space but the received rewards are zero. To facilitate the model's learning of this behavior, we (1) relabel the observations, available actions, and continuation flags of absent agents within an episode, and (2) append null observations, available actions, zero rewards, and zero continuation flags to the end of the trajectory. This approach enables the latent dynamics model to effectively learn and represent agent absence.

### 4.3 LEARNING DECENTRALIZED POLICIES IN LATENT SPACE

The disentangled latent states learned by our model enable effective training of decentralized policies through imagination. The multi-agent dynamics of DMAWM effectively forms a factored n-agent Dec-MDP (Bernstein et al., 2002) where the joint latent state $I_t^{1:n} = (h_t^i, z_t^i)_{i=1}^n$ uniquely determines the underlying state (see Appendix A.1).

**Generating multi-agent trajectories in imagination** The imaginary rollouts begin by encoding a subsequence of a ground-truth trajectory $\tau_H^{1:n} = (o_{1:H}^{1:n}, a_{1:H-1}^{1:n})$ using the agent module, where $H$ is the length of the subsequence. The resulting joint latent state $(h_1^{1:n}, z_1^{1:n})$ is used as the initial state for imagination. Then for each agent $i$, the actor model selects the action $a_t^i \sim \pi_\theta(\cdot \mid h_t^i, z_t^i)$ and the recurrent model updates the deterministic state $h_t^i$. After that, the interaction predictor samples the joint stochastic state $z_t^{1:n} \sim p_\phi(\cdot \mid h_t^{1:n})$ for all agents. This process is repeated for $L$ imagination steps, resulting in an imaginary trajectory. After an imagination rollout, the other components of the trajectory are generated by the decoders for training, including the reward $\hat{r}_t^i$, continuation flag $\hat{c}_t^i$, and available actions $\hat{m}_t^i$.

**Actor-critic loss** We train the actor and critic similar to MAPPO (Yu et al., 2022) but entirely using the imaginary trajectories. To accurately estimate the value, the centralized critic $v_\xi^{1:n}(h_t^{1:n})$ utilizes a transformer to contextualize the joint deterministic state and estimates the value for each agent. We also bootstrap the value at the next step and calculate the advantage via the generalized advantage estimator (GAE) (Schulman et al., 2021) to balance the bias and variance:

$$A_t^i = \delta_t^i + \lambda\gamma\hat{c}_{t+1}^i A_{t+1}^i, \qquad \delta_t^i = \hat{r}_t + \gamma\hat{c}_{t+1}^i v_\xi^i(h_{t+1}^{1:n}) - v_\xi^i(h_t^{1:n}), \tag{7}$$

where $\lambda$ balances the bias and variance for advantage estimation, and $\hat{c}_{t+1}^i$ is the continuation flag at the next step. We compute advantages and value estimates for all agents along the trajectory. The actor and critic losses are

$$\mathcal{L}_{\text{actor}}(\theta) = -\sum_{t=1}^{L}\sum_{i=1}^{n} \min(r_t^i(\theta)A_t^i, \text{clip}(r_t^i(\theta), 1-\epsilon, 1+\epsilon)A_t^i) - \beta_{\text{ent}}\mathcal{H}(\pi_\theta(\cdot \mid h_t^i, z_t^i)), \tag{8}$$

$$\mathcal{L}_{\text{critic}}(\xi) = \sum_{t=1}^{L}\sum_{i=1}^{n} (\text{sg}(v_\xi^i(h_t^{1:n}) + A_t^i) - v_\xi^i(h_t^{1:n}))^2, \tag{9}$$

where $r_t^i(\theta) = \frac{\pi_\theta(a_t^i \mid h_t^i, z_t^i)}{\pi_{\theta\text{old}}(a_t^i \mid h_t^i, z_t^i)}$ is the ratio of the current policy to the previous policy, the clipping operation constrains the ratio to lie within $[1-\epsilon, 1+\epsilon]$, $\mathcal{H}(\pi_\theta(\cdot \mid h_t^i, z_t^i))$ is the policy entropy, $\beta_{\text{ent}}$ is the entropy coefficient, and $v_\xi^i(h_t^{1:n}) + A_t^i$ is the value target for agent $i$ at time $t$.

## 5 EXPERIMENTS

In this section, we conduct experiments to demonstrate the effectiveness of DMAWM on three MARL benchmarks, comparing it against both model-based and model-free baselines. To analyze its interaction modeling capability, we visualize the generated imaginary trajectories. We also conduct a wall-clock-time comparison to evaluate the runtime efficiency of DMAWM in Appendix A.7.

### 5.1 EXPERIMENTAL SETUP

**Benchmarks** We evaluate our method on three MARL benchmarks that pose complementary challenges: SMAC (Samvelyan et al., 2019) features complex dynamics and diverse coordination patterns; SMACv2 (Ellis et al., 2023), an updated version of SMAC, introduces greater randomness in

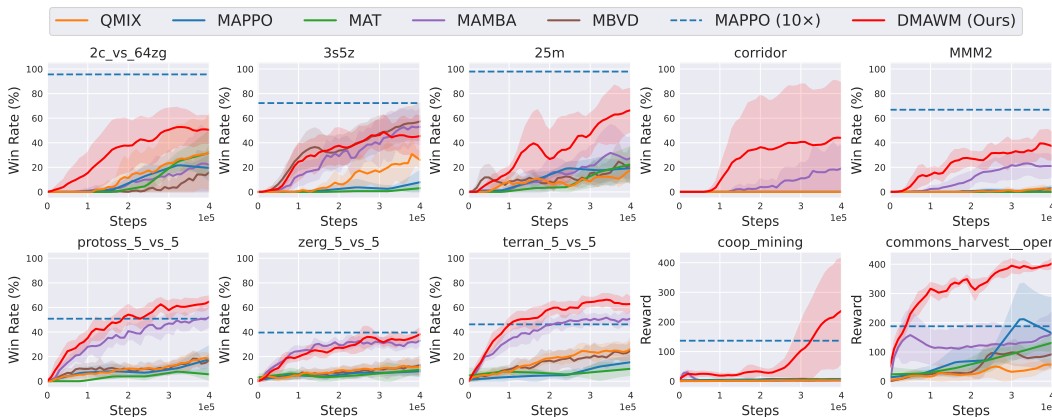

Figure 2: Training curves comparing DMAWM with model-based and model-free baselines on three MARL benchmarks: SMAC, SMACv2, and Melting Pot. Results are averaged over 5 independent runs, with shaded areas representing the standard deviation. All algorithms are trained for 400K environment steps. Dashed lines show performance of MAPPO after 10 times more environment steps than model-based algorithms.

the starting positions and unit types; Melting Pot (Leibo et al., 2021) uses visual observations and requires behavior switching according to the context. We train all algorithms for 400K environment steps on each benchmark. A detailed description of the benchmarks is provided in Appendix A.5.

**Baselines** We compare our method against both model-based and model-free baselines. The model-based approaches include the communication-free MBVD (Xu et al., 2022) which reconstructs observations to train a value decomposition method; and the communication-based MAMBA (Egorov & Shpilman, 2022), a current SOTA method based on DreamerV2. The model-free baselines include QMIX (Rashid et al., 2018), MAPPO (Yu et al., 2022), and MAT (Wen et al., 2022). For QMIX, we utilize the tuned implementation from (Hu et al., 2021), which has demonstrated competitive performance across various benchmarks.

**Implementation details** We perform one training step of both the world model and the policies every 32 environment steps, beginning after an initial 5000-step warm-up. During training, the learned model generates imaginary trajectories of 16 steps with 1024 parallel rollouts. For Melting Pot, which uses visual observations, inputs are downsampled from $88 \times 88$ to $44 \times 44$ pixels to reduce GPU memory usage. Visual observations are encoded with a CNN and decoded with a transposed CNN (Dumoulin & Visin, 2016). To improve sample efficiency, we share parameters of the agent modules and policies across all agents. To ensure fairness for comparison, all algorithms use the same set of hyperparameters across benchmarks. Additional implementation details and hyperparameters are provided in Appendix A.3 and Appendix A.4, respectively.

## 5.2 PERFORMANCE COMPARISON

Table 1 and Figure 2 summarize the performance comparison. DMAWM demonstrates strong performance across all benchmarks. Across benchmarks, DMAWM is consistently sample-efficient, matching or surpassing the performance of the strong baselines.

On SMAC, DMAWM learns substantially faster and attains the highest win rates compared to baselines, especially on the 2c_vs_64zg and corridor maps. Notably, the 25m map features significantly more agents than the others, which poses great challenges to model-based methods. Nevertheless, DMAWM has shown great scalability compared to the other model-based baselines. In the more stochastic SMACv2 environment, both DMAWM and MAMBA outperform other baselines, with DMAWM achieving slightly better final performance. For the vision-based scenarios from Melting Pot, DMAWM obtains the highest returns without specific tuning, highlighting its effectiveness in modeling agent interactions even in the visual domain. Its efficacy in modeling these interactions is further shown via latent state visualizations in Section 5.4.

Table 1: Performance comparison across SMAC, SMACv2, and Melting Pot benchmarks. We compared our approach against both model-based and model-free baselines. Evaluation metrics are win rate (%) for SMAC and SMACv2, and episode return for Melting Pot. All results are reported as the average over 5 independent runs, accompanied by their standard deviations.

| Benchmarks | Maps | Model-free | | | Model-based | | |
|---|---|---|---|---|---|---|---|
| | | QMIX | MAPPO | MAT | MAMBA | MBVD | DMAWM (Ours) |
| SMAC | 2c_vs_64zg | 31.8 (26.1) | 19.8 (7.7) | 30.9 (14.3) | 22.8 (25.0) | 14.8 (18.3) | **52.1** (9.4) |
| | 3s5z | 28.9 (8.8) | 7.2 (7.4) | 2.7 (3.5) | 53.2 (12.3) | **59.2** (16.1) | 45.1 (15.2) |
| | 25m | 16.6 (7.2) | 19.1 (7.1) | 21.6 (12.2) | 24.5 (12.7) | 21.2 (11.1) | **71.1** (9.3) |
| | corridor | 0.0 (0.0) | 0.0 (0.0) | 0.0 (0.0) | 18.6 (17.7) | 0.0 (0.0) | **45.2** (39.2) |
| | MMM2 | 2.3 (2.3) | 2.7 (1.7) | 0.0 (0.0) | 21.2 (8.1) | 0.2 (0.7) | **37.6** (10.8) |
| SMACv2 | protoss_5_vs_5 | 19.0 (4.0) | 16.9 (9.2) | 5.9 (3.8) | 51.4 (8.3) | 14.9 (6.3) | **64.8** (4.9) |
| | zerg_5_vs_5 | 11.3 (5.9) | 7.9 (3.6) | 9.0 (3.8) | 32.4 (4.0) | 12.7 (6.5) | **38.7** (4.9) |
| | terran_5_vs_5 | 25.6 (8.4) | 15.0 (4.9) | 9.8 (4.7) | 50.9 (5.3) | 24.1 (7.7) | **62.3** (6.2) |
| Melting Pot | Coop Mining | 2.8 (2.4) | 6.6 (2.7) | 4.9 (1.0) | 2.6 (2.1) | 5.7 (2.6) | **244.2** (150.6) |
| | Commons Harvest: Open | 61.5 (13.4) | 170.6 (105.2) | 127.6 (23.5) | 175.9 (69.4) | 105.0 (30.1) | **401.7** (16.6) |

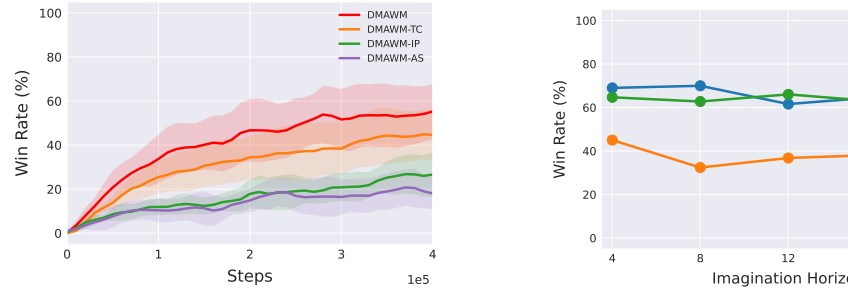

(a) Ablation on core components      (b) Ablation on imagination horizon

Figure 3: Ablation study on SMACv2. (a) We remove each of the core components to study the individual contributions, and the results are averaged across the 3 SMACv2 maps. (b) We study the impact of the imagination horizon on the performance of DMAWM. Each map is trained for 400K environment steps with 3 independent runs. The results underscore the importance of DMAWM's core components and demonstrate its robustness to the choice of imagination horizon.

## 5.3 ABLATION STUDY

To assess the individual contributions of DMAWM's core components—the transformer-based critic, interaction predictor, and absorbing state—we conduct ablation studies on them. To ablate the transformer-based critic, we replace it with an MLP that estimates the value based on the latent states of an agent. We call this ablation DMAWM-TC. The second ablation, DMAWM-IP, replaces the interaction predictor with an MLP, wherein each agent independently predicts its own latent state and other trajectory components. The third ablation, DMAWM-AS, removes the absorbing state mechanism. In this setup, agent absence is predicted using a binary value for each agent, and absent agents are no longer considered in future trajectory predictions. As shown in Figure 3a, all ablations perform worse than the full DMAWM, underscoring the importance of these components.

On figure 3b, we also study the impact of the imagination horizon on the performance of DMAWM, and we find that the performance of DMAWM is robust to the imagination horizon.

## 5.4 VISUALIZATION OF THE LATENT SPACE

To qualitatively evaluate the multi-agent latent dynamics model's ability to capture agent interactions, we generate an imaginary trajectory and compare it with the ground-truth trajectory, as shown in Figure 4. The figure displays the decoded observations of two out of the four agents within the same trajectory. We use the representation model to encode the initial 6 frames and generate the subsequent 24 frames with the interaction predictor and the observation decoder.

From the visualization, we can see that the positions of agents and the structure of the wall in the imagination align well with the real environment. While the ore distribution in imagination (iron

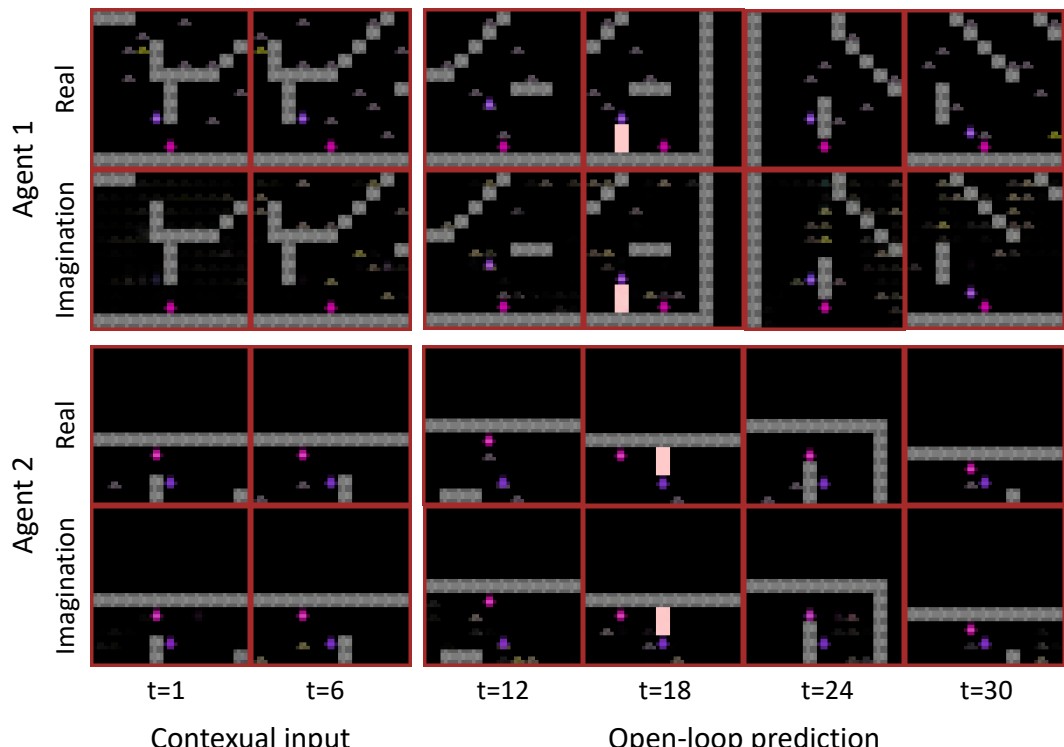

Figure 4: Long-horizon trajectory prediction by our multi-agent latent dynamics model in the Coop Mining environment. Conditioned on an initial 6-frame context from a hold-out trajectory and corresponding actions, the model employs the interaction predictor to generate 24 subsequent frames in the latent space. We selectively show the decoded observations for two out of the four agents, to demonstrate the model's ability to capture multi-agent interactions coherently over long horizons.

ore is marked in gray and gold ore is marked in yellow) aligns more closely with the ground-truth trajectory at early stage, they diverge at later timesteps. This is expected since the models must infer ore locations in unobserved areas. What we find interesting is that the relative positions of the two agents in imagination remain consistent with each other in their ego-centric observations throughout all timesteps. Notably, at $t = 18$, agent 2's action (mining beam) is accurately reflected in agent 1's decoded observation, highlighting the model's efficacy in capturing agent interactions.

## 6 CONCLUSION AND FUTURE WORK

In this work, we addressed the critical challenge of learning effective decentralized policies for multi-agent tasks using world models. We introduced the Disentangled Multi-Agent World Model (DMAWM), a framework that learns decentralized policies in the latent space with a novel architecture featuring independent agent modules and a shared environment module. This architecture enables it to learn a factorized latent representation that explicitly captures agent interactions while effectively disentangling individual agent latent states. This disentanglement is crucial for training decentralized policies via imagined trajectories. Our experiments on challenging MARL benchmarks, with both vector and visual observations, demonstrated that DMAWM significantly outperforms existing model-based and model-free baselines in sample efficiency and final performance.

While this paper mainly focuses on cooperative tasks, extending DMAWM to mixed-motive scenarios is a promising direction, as the core mechanisms are not inherently tied to cooperative tasks. We also find that the multi-agent world model tends to overfit to the trajectories generated by the trained policies. Investigating methods to enhance the world model's quality by promoting policy diversity during training could lead to more robust and generalizable world models capable of generating more realistic trajectories that reflect the real environment.

REPRODUCIBILITY STATEMENT

We provide detailed information to reproduce our results. The DMAWM architecture, training objectives, and training procedure are described in Section 4.1, Section 4.2, and Section 4.3. Experimental setups, and benchmarks are presented in Section 5.1, with other implementation details can be found in Appendix A.3. An introduction of the benchmarks we use is provided in Appendix A.5. The full hyperparameter table can be found in Appendix A.4. Appendix A.2 contains the complete derivation of the training objective.

We also submit anonymized source code as supplementary material, which includes the scripts to reproduce the results in the main experiments.

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

# A APPENDIX

## A.1 CONNECTIONS BETWEEN MULTI-AGENT LATENT DYNAMICS AND TRADITIONAL MULTI-AGENT FORMULATIONS

In this section, we build the connections between the multi-agent latent dynamics and traditional multi-agent formulations, hopefully the techniques developed for these specific domains can be applied to our framework.

The multi-agent latent dynamics model can be seen as an action-conditioned generative model of the joint observation sequence that can be factorized as:

$$
\begin{aligned}
p(o_{1:T}^{1:n}, I_{1:T}^{1:n} \mid a_{1:T-1}^{1:n}) &= p(I_{1:T}^{1:n} \mid a_{1:T-1}^{1:n}) p(o_{1:T}^{1:n} \mid I_{1:T}^{1:n}) \\
&= \prod_{t=1}^{T} p(I_t^{1:n} \mid I_{t-1}^{1:n}, a_{t-1}^{1:n}) p(o_t^{1:n} \mid I_t^{1:n}) \\
&= \prod_{t=1}^{T} p(I_t^{1:n} \mid I_{t-1}^{1:n}, a_{t-1}^{1:n}) \prod_{i=1}^{n} p(o_t^i \mid I_t^i),
\end{aligned}
\tag{10}
$$

where $p(I_t^{1:n} \mid I_{t-1}^{1:n}, a_{t-1}^{1:n})$ is the transition probability and $p(o_t^i \mid I_t^i)$ is the observation probability.

To formulate the interaction derived from the above multi-agent latent dynamics model, we can define the state as the collection of the individual latent states of all agents $s_t = (I_t^1, \ldots, I_t^n)$ as the sequence $(I_t^{1:n})_{t=1}^T$ is Markovian on each of its components. The transition probability is governed by the multi-agent latent dynamics model $p(s_t \mid s_{t-1}, a_{t-1}^{1:n}) = p(I_t^{1:n} \mid I_{t-1}^{1:n}, a_{t-1}^{1:n})$. Each agent could observe its component of the joint latent state $o_t^i = I_t^i$, leading to the fact that the joint observation is equivalent to the state, i.e., $o_t^{1:n} = I_t^{1:n} = s_t$. This interaction formulation is captured by an *agent-wise factored Dec-MDP* (Goldman & Zilberstein, 2004), where the state is uniquely determined by the joint observation of all agents.

## A.2 DERIVATION OF THE ELBO

Here we derive the evidence lower bound (ELBO) for the joint observation sequence $o_{1:T}^{1:n}$ given the action sequence $a_{1:T-1}^{1:n}$. The derivation for the other trajectory components are similar.

The posterior is the representation model that updates recursively:

$$
q(I_{1:T}^{1:n} \mid a_{1:T-1}^{1:n}, o_{1:T}^{1:n}) = \prod_{t=1}^{T} q(I_t^{1:n} \mid I_{t-1}^{1:n}, a_{t-1}^{1:n}, o_t^{1:n}).
\tag{11}
$$

We also need the marginal posterior to derive the ELBO:

$$
\begin{aligned}
\sum_{I_{t \neq l}^{1:n}} q(I_{1:T}^{1:n} \mid a_{1:T-1}^{1:n}, o_{1:T}^{1:n}) &= \sum_{I_{1:l-1}^{1:n}} \sum_{I_{l+1:T}^{1:n}} q(I_{1:T}^{1:n} \mid a_{1:T-1}^{1:n}, o_{1:T}^{1:n}) \\
&= \sum_{I_{1:l-1}^{1:n}} \sum_{I_{l+1:T}^{1:n}} \prod_{t=1}^{T} q(I_t^{1:n} \mid I_{t-1}^{1:n}, a_{t-1}^{1:n}, o_t^{1:n}) \qquad \text{Apply Eq. 11} \\
&= \sum_{I_{1:l-1}^{1:n}} \sum_{I_{l+1:T}^{1:n}} \prod_{t=1}^{l} q(I_t^{1:n} \mid I_{t-1}^{1:n}, a_{t-1}^{1:n}, o_t^{1:n}) \prod_{t=l+1}^{T} q(I_t^{1:n} \mid I_{t-1}^{1:n}, a_{t-1}^{1:n}, o_t^{1:n}) \\
&= \sum_{I_{1:l-1}^{1:n}} \underbrace{\prod_{t=1}^{l} q(I_t^{1:n} \mid I_{t-1}^{1:n}, a_{t-1}^{1:n}, o_t^{1:n})}_{=q(I_{1:l}^{1:n} \mid a_{1:l-1}^{1:n}, o_{1:l}^{1:n}) \text{ by Eq. 11}} \underbrace{\sum_{I_{l+1:T}^{1:n}} \prod_{t=l+1}^{T} q(I_t^{1:n} \mid I_{t-1}^{1:n}, a_{t-1}^{1:n}, o_t^{1:n})}_{=1} \\
&= \sum_{I_{1:l-1}^{1:n}} q(I_{1:l}^{1:n} \mid a_{1:l-1}^{1:n}, o_{1:l}^{1:n}) = q(I_l^{1:n} \mid a_{1:l-1}^{1:n}, o_{1:l}^{1:n}).
\end{aligned}
\tag{12}
$$

The ELBO is derived as follows:

$$\ln p(o_{1:T}^{1:n} \mid a_{1:T-1}^{1:n})$$

$$= \ln \sum_{I_{1:T}^{1:n}} p(o_{1:T}^{1:n}, I_{1:T}^{1:n} \mid a_{1:T-1}^{1:n})$$

$$= \ln \sum_{I_{1:T}^{1:n}} p(I_{1:T}^{1:n} \mid a_{1:T-1}^{1:n}) p(o_{1:T}^{1:n} \mid I_{1:T}^{1:n}) \qquad \text{Apply Eq. 10}$$

$$= \ln \sum_{I_{1:T}^{1:n}} q(I_{1:T}^{1:n} \mid a_{1:T-1}^{1:n}, o_{1:T}^{1:n}) \frac{p(I_{1:T}^{1:n} \mid a_{1:T-1}^{1:n})}{q(I_{1:T}^{1:n} \mid a_{1:T-1}^{1:n}, o_{1:T}^{1:n})} p(o_{1:T}^{1:n} \mid I_{1:T}^{1:n})$$

$$= \ln \mathbb{E}_{q(I_{1:T}^{1:n} \mid a_{1:T-1}^{1:n}, o_{1:T}^{1:n})} \left[ \frac{p(I_{1:T}^{1:n} \mid a_{1:T-1}^{1:n})}{q(I_{1:T}^{1:n} \mid a_{1:T-1}^{1:n}, o_{1:T}^{1:n})} p(o_{1:T}^{1:n} \mid I_{1:T}^{1:n}) \right]$$

$$\geq \mathbb{E}_{q(I_{1:T}^{1:n} \mid a_{1:T-1}^{1:n}, o_{1:T}^{1:n})} \left[ \ln \frac{p(I_{1:T}^{1:n} \mid a_{1:T-1}^{1:n})}{q(I_{1:T}^{1:n} \mid a_{1:T-1}^{1:n}, o_{1:T}^{1:n})} p(o_{1:T}^{1:n} \mid I_{1:T}^{1:n}) \right] \qquad \text{Jensen's inequality}$$

$$= \mathbb{E}_{q(I_{1:T}^{1:n} \mid a_{1:T-1}^{1:n}, o_{1:T}^{1:n})} \left[ \sum_{t=1}^{T} \ln p(o_t^{1:n} \mid I_t^{1:n}) - \sum_{t=1}^{T} \ln \frac{q(I_t^{1:n} \mid I_{t-1}^{1:n}, a_{t-1}^{1:n}, o_t^{1:n})}{p(I_t^{1:n} \mid I_{t-1}^{1:n}, a_{t-1}^{1:n})} \right]$$

$$= \sum_{t=1}^{T} \mathbb{E}_{q(I_{1:T}^{1:n} \mid a_{1:T-1}^{1:n}, o_{1:T}^{1:n})} \left[ \ln p(o_t^{1:n} \mid I_t^{1:n}) \right] - \sum_{t=1}^{T} \mathbb{E}_{q(I_{1:T}^{1:n} \mid a_{1:T-1}^{1:n}, o_{1:T}^{1:n})} \left[ \ln \frac{q(I_t^{1:n} \mid I_{t-1}^{1:n}, a_{t-1}^{1:n}, o_t^{1:n})}{p(I_t^{1:n} \mid I_{t-1}^{1:n}, a_{t-1}^{1:n})} \right]$$

$$= \sum_{t=1}^{T} \underbrace{\mathbb{E}_{q(I_{1:t}^{1:n} \mid o_{1:t}^{1:n}, a_{1:t-1}^{1:n})}}_{\text{marginalized by Eq. 12}} \left[ \ln p(o_t^{1:n} \mid I_t^{1:n}) \right]$$

$$- \sum_{t=1}^{T} \underbrace{\mathbb{E}_{q(I_{t-1}^{1:n} \mid o_{1:t-1}^{1:n}, a_{1:t-2}^{1:n})}}_{\text{marginalized by Eq. 12}} \left[ D_{\mathrm{KL}} \left( q(\cdot \mid I_{t-1}^{1:n}, a_{t-1}^{1:n}, o_t^{1:n}) \, \| \, p(\cdot \mid I_{t-1}^{1:n}, a_{t-1}^{1:n}) \right) \right]$$

$$= \sum_{t=1}^{T} \mathbb{E}_{q(I_t^{1:n} \mid o_{1:t}^{1:n}, a_{1:t-1}^{1:n})} \left[ \sum_{i=1}^{n} \ln p(o_t^i \mid I_t^i) \right]$$

$$- \sum_{t=1}^{T} \mathbb{E}_{q(I_{t-1}^{1:n} \mid o_{1:t-1}^{1:n}, a_{1:t-2}^{1:n})} \left[ D_{\mathrm{KL}} \left( \prod_{i=1}^{n} q(\cdot \mid I_{t-1}^i, a_{t-1}^i, o_t^i) \, \| \, p(\cdot \mid I_{t-1}^{1:n}, a_{t-1}^{1:n}) \right) \right].$$

### A.3 OTHER IMPLEMENTATION DETAILS

The interaction predictor is implemented using a Transformer network (Vaswani et al., 2017). It is tasked with predicting the discrete latent state, conditioned on the joint deterministic state $p_\phi(z_t^{1:n} \mid h_t^{1:n})$. Each discrete latent state is represented by 32 one-hot vectors, each with 32 classes. The prediction process begins by encoding the joint deterministic state $h_t^{1:n}$ using a Transformer encoder, which yields $(\bar{h}_t^1, \ldots, \bar{h}_t^n) = \text{TransformerEncoder}(h_t^1, \ldots, h_t^n)$. Subsequently, a Multi-Layer Perceptron (MLP) maps each resulting embedding $\bar{h}_t^i$ to a 1024-dimensional vector. This vector is then reshaped to facilitate the sampling of the discrete latent state.

The reward decoder $p_\phi(\hat{r}_t^{1:n} \mid h_t^{1:n})$, continuation decoder $p_\phi(\hat{c}_t^{1:n} \mid h_t^{1:n})$, and available actions decoder $p_\phi(\hat{m}_t^{1:n} \mid h_t^{1:n})$ are implemented with a shared Transformer network. First, the shared Transformer encoder blocks processes the joint deterministic state $h_t^{1:n}$, producing $(\tilde{h}_t^1, \ldots, \tilde{h}_t^n) = \text{TransformerEncoder}(h_t^1, \ldots, h_t^n)$. Then, each decoder employs a separate MLP head to produce its respective output. For the centralized critic, we use a separate Transformer encoder to encode the joint deterministic state $h_t^{1:n}$, then map the embedding to a value estimate for each agent.

We also adopt tricks from Dreamerv3, such as the symexp twohot loss for the reward decoder and critic, and free bits for the dynamics loss and representation loss. The reward decoder and critic use the symexp twohot loss as Dreamerv3 (Hafner et al., 2025). To be specific, the out-

puts of reward decoder and critic can be represented as the weighted average of exponentially spaced bins, e.g., $\hat{r}_t^i = \text{Softmax}(\text{MLP}(h_t^i, z_t^i))^\top B$ where $B = \text{symexp}(-20, \ldots, +20)$ and $\text{symexp}(x) = \text{sign}(x)(\exp(|x|-1))$. The reward decoder and critic are trained to match the two-hot target using cross-entropy loss.

## A.4 HYPERPARAMETERS

### A.4.1 HYPERPARAMETERS FOR DMAWM

The empirical results of our DMAWM implementation is based on the hyperparameters in Table 2.

Table 2: Hyperparameters for the DMAWM algorithm.

| Hyperparameter | Value |
|---|---|
| **Reinforcement Learning** | |
| Optimizer | Adam |
| Entropy coefficient | 0.01 |
| PPO epochs | 5 |
| Clip param | 0.2 |
| Actor learning rate | $3 \times 10^{-5}$ |
| Critic learning rate | $3 \times 10^{-5}$ |
| Discount factor | 0.99 |
| GAE lambda | 0.95 |
| **World Model** | |
| Max grad norm | 100 |
| Model learning rate | $1 \times 10^{-4}$ |
| Model batch size | 16 |
| Sequence length | 64 |
| Rollout horizon | 16 |
| Buffer size | $2.5 \times 10^5$ |
| KL balancing entropy weight | 0.2 |
| KL balancing cross entropy weight | 0.8 |
| Discrete latent dimensions | 32 |
| Discrete latent classes | 32 |
| Transformer layers | 3 |
| Transformer heads | 8 |
| Decoder hidden size | 1024 |
| Decoder layers | 2 |

### A.4.2 HYPERPARAMETERS FOR BASELINES

The experimental results on MAPPO (Yu et al., 2022) is based on the official implementation[2] with the following hyperparameters in Table 3.

The experimental results on MAT (Yu et al., 2022) is based on the implementation[3] with the following hyperparameters in Table 4.

The experimental results on QMIX (Rashid et al., 2018) is based on the optimized implementation of PyMARL2[4] with the following hyperparameters in Table 5.

The experimental results on MAMBA (Egorov & Shpilman, 2022) is based on the official implementation[5] with the following hyperparameters in Table 6.

---

[2]https://github.com/marlbenchmark/on-policy
[3]https://github.com/marlbenchmark/on-policy
[4]https://github.com/hijkzzz/pymarl2
[5]https://github.com/jbr-ai-labs/mamba

Table 3: Hyperparameters for the MAPPO algorithm.

| Hyperparameter | Value |
|---|---|
| Use RNN | True |
| Optimizer | Adam |
| Episode length | 400 |
| Entropy coefficient | 0.01 |
| Discount factor | 0.99 |
| GAE lambda | 0.95 |
| Critic learning rate | $5 \times 10^{-4}$ |
| Actor learning rate | $5 \times 10^{-4}$ |
| PPO epochs | 5 |
| Clip param | 0.2 |
| Parallel workers | 8 |
| Max grad norm | 10 |

Table 4: Hyperparameters for the MAT algorithm.

| Hyperparameter | Value |
|---|---|
| Use RNN | True |
| Optimizer | Adam |
| Episode length | 400 |
| Entropy coefficient | 0.01 |
| Discount factor | 0.99 |
| GAE lambda | 0.95 |
| Critic learning rate | $5 \times 10^{-4}$ |
| Actor learning rate | $5 \times 10^{-4}$ |
| PPO epochs | 5 |
| Clip param | 0.2 |
| Parallel workers | 8 |
| Max grad norm | 10 |
| Transformer layers | 1 |
| Transformer heads | 1 |

Table 5: Hyperparameters for the QMIX algorithm.

| Hyperparameter | Value |
|---|---|
| Use RNN | True |
| Optimizer | Adam |
| Learning rate | 0.001 |
| Discount factor | 0.99 |
| Target update interval (episodes) | 200 |
| Max grad norm | 10 |
| Batch size | 128 |
| Buffer size (episodes) | 5000 |
| Epsilon | $1.00 \rightarrow 0.05$ |
| TD lambda | 0.6 |

Table 6: Hyperparameters for the MAMBA algorithm.

| Hyperparameter | Value |
|---|---|
| **Reinforcement Learning** | |
| Optimizer | Adam |
| Entropy coefficient | 0.001 |
| Number of updates | 4 |
| PPO epochs | 5 |
| Clip param | 0.2 |
| Actor learning rate | $5 \times 10^{-4}$ |
| Critic learning rate | $5 \times 10^{-4}$ |
| Discount factor | 0.99 |
| GAE lambda | 0.95 |
| **World Model** | |
| Model learning rate | $2 \times 10^{-4}$ |
| Model epochs | 60 |
| Model batch size | 40 |
| Sequence length | 20 |
| Rollout horizon | 15 |
| Buffer size | $2.5 \times 10^{5}$ |
| KL balancing entropy weight | 0.2 |
| KL balancing cross entropy weight | 0.8 |
| Max grad norm | 100 |
| Trajectories between updates | 1 |

The experimental results on MBVD (Xu et al., 2022) is based on the official implementation submitted to OpenReview[6] with the following hyperparameters in Table 7.

Table 7: Hyperparameters for the MBVD algorithm.

| Hyperparameter | Value |
|---|---|
| **Reinforcement Learning** | |
| Optimizer | RMSProp |
| Learning rate | $5 \times 10^{-4}$ |
| Discount factor | 0.99 |
| Target update interval (episodes) | 200 |
| Max grad norm | 10 |
| Batch size | 32 |
| Buffer size (episodes) | 5000 |
| Epsilon | $1.00 \rightarrow 0.05$ |
| **World Model** | |
| Rollout horizon | 3 |
| KL balancing entropy weight | 0.3 |
| KL balancing cross entropy weight | 0.7 |
| Trajectories between updates | 1 |

## A.5 ENVIRONMENT DESCRIPTIONS

### A.5.1 SMAC

StarCraft Multi-Agent Challenge (SMAC) (Samvelyan et al., 2019) is a popular benchmark for MARL research based on the real-time strategy game StarCraft II. It offers a collection of micro

---

[6]https://openreview.net/forum?id=flBYpZkW6ST

battle scenarios in StarCraft II, where a team of ally units must collaborate to defeat the opposing team controlled by rule-based bots. In these scenarios, each agent is responsible for controlling one ally unit and has access to information such as the distance, relative location, health, shield, and type of both ally and enemy units within their field of vision. For our purposes, we consider each unit as an entity, with ally units categorized as agent entities and enemies as non-agent entities.

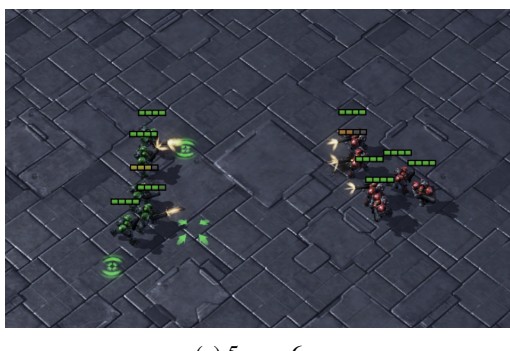 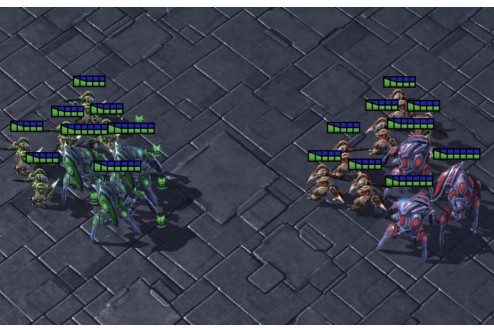

(a) 5m vs 6m                                    (b) 3s5z_vs_3s6z

Figure 5: SMAC

### A.5.2 SMACv2

SMACv2 (Ellis et al., 2023) extends SMAC by introducing increased complexity and randomness. It randomizes the starting positions and unit types of agents with varying sight and attack ranges, presenting MARL algorithms with greater levels of stochasticity and diversity. Similar to the approach taken in SMAC, we consider each unit as an individual entity, ally units as agent entities, and enemies as non-agent entities.

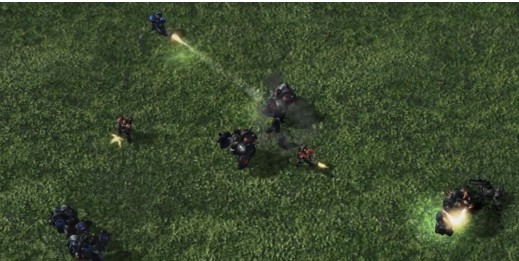

Figure 6: SMACv2

### A.5.3 MELTING POT

Melting Pot (Leibo et al., 2021) provides a suite of multi-agent tasks and an evaluation protocol for assessing the social intelligence of agents. These tasks are vision-based, where the observations are ego-centric 2D visual observations of the environment.

Coop Mining (Figure 7a), a cooperative scenario where agents coordinate to collect resources, is an instance of the cooperative task in Melting Pot. The environment features two resource types: iron ore and gold ore. Iron ore can be gathered by a single agent, but gold ore necessitates the use of beams by two agents within a time window of 3 timesteps. Collecting iron ore yields a reward of 1 for the agent, while successfully gathering gold ore grants a reward of 8 to each participating agent. Each episode lasts 1000 timesteps.

Common Harvest (Figure 7b), in which agents consume renewable common resources, is a tragedy-of-the-commons scenario. Apples are initially scattered throughout the environment, and consuming one yields a reward of 1. At each timestep, apples respawn with a probability that is positively correlated with the number of apples within a neighborhood of radius 2. Consequently, an isolated patch (one with no other apples within distance 2) can be permanently depleted if all apples in

that patch are consumed. Agents must therefore exercise restraint when consuming apples within a patch. Each episode lasts 1000 timesteps.

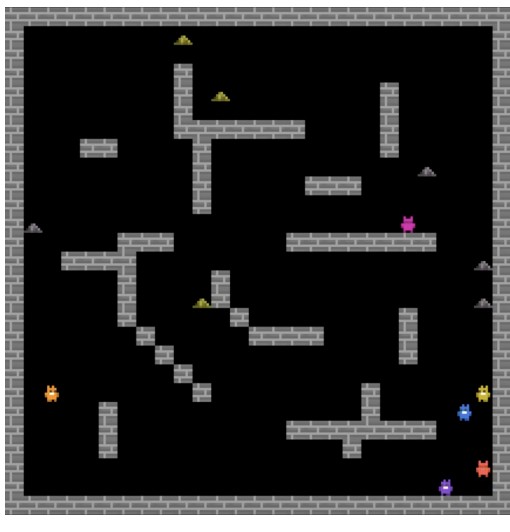

(a) Coop Mining

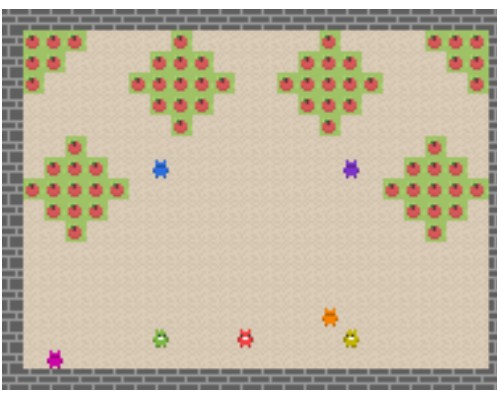

(b) Common Harvest

Figure 7: Melting Pot

To save GPU memory, we resize the observation from $88 \times 88$ to $44 \times 44$. We show an example of the original and resized observations in Figure 8 below.

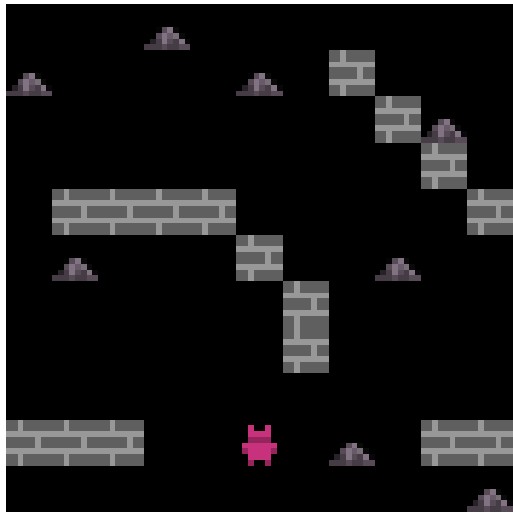

(a) Original observation

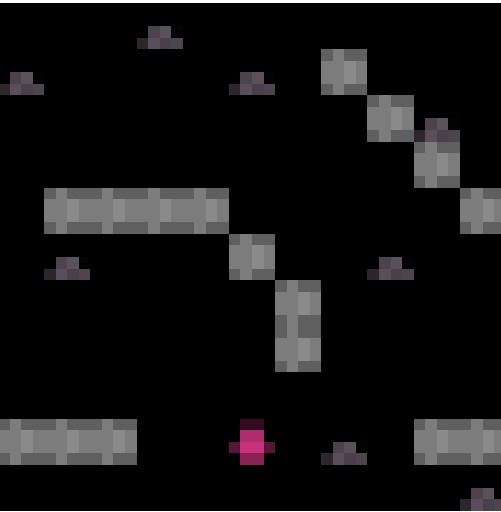

(b) Resized observation

Figure 8: Observation Resizing

## A.6 COMPUTATIONAL RESOURCES

Most experiments were conducted with NVIDIA RTX 3090 GPUs. The time for an experiment is highly dependent on the number of agents and total timesteps. For example, the 2s_vs_1sc map of the SMAC benchmark, which has 2 agents and the total timesteps is 400K, takes around 12 hours to train.

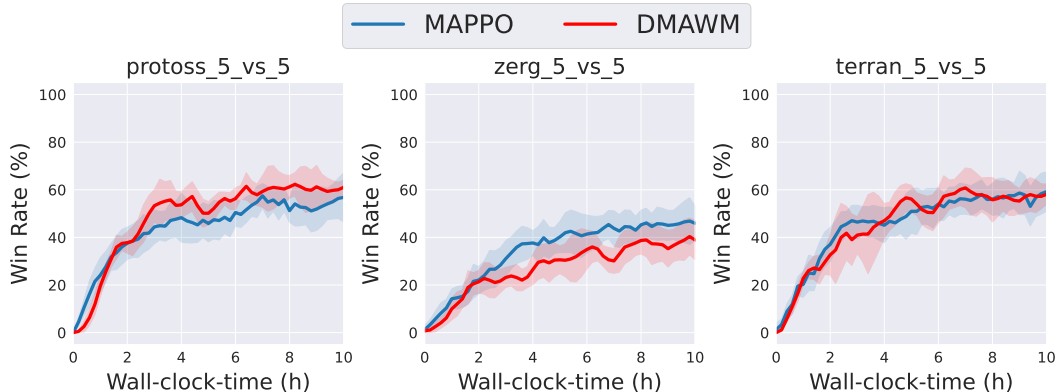

Figure 9: Wall-clock-time comparison between DMAWM and MAPPO on the SMACv2 benchmark. For DMAWM, we set the imagination horizon to 4. Results are averaged over 3 independent runs.

### A.7 WALL-CLOCK-TIME COMPARISON

To evaluate the runtime efficiency of DMAWM, we compare the wall-clock-time of DMAWM against MAPPO on the SMACv2 benchmark. For DMAWM, we set the imagination horizon to 4. Both algorithms are trained under the same computational resources. As shown in Figure 9, DMAWM achieves comparable training speed as MAPPO while using much less environment steps to reach the same performance. This result is particularly interesting to us, as it effectively closes the runtime gap between model-based and model-free algorithms.

