# OpenReview forum: "Learning Disentangled Multi-Agent World Model for Decentralized Control"
_ICLR.cc/2026/Conference — ICLR 2026 Conference Withdrawn Submission_

### Official Review · Reviewer_JXMY · 2025-10-27

**Soundness:** 3
**Presentation:** 3
**Contribution:** 2
**Rating:** 4
**Confidence:** 5

**Summary:**

This paper addresses the challenge of applying world models to decentralized multi-agent settings. The authors propose the Disentangled Multi-Agent World Model (DMAWM), a framework designed to learn factorized latent representations. The architecture consists of independent agent modules that process local observations and a shared environment module that models their interactions during imagination. By training the environment module's joint prior to match the factorized posterior from the agent modules, the method aims to learn disentangled latent states. The effectiveness of DMAWM is demonstrated through experiments on SMAC, SMACv2, and Melting Pot benchmarks, where it outperforms existing model-based and model-free methods in sample efficiency and performance.

**Strengths:**

1.  The paper tackles an important problem: effectively training decentralized multi-agent policies in a sample-efficient manner using world models.
2.  The proposed architecture, which separates independent agent modules from a shared environment module, is an intuitive approach to learning factorized representations and avoiding spurious correlations.
3.  The proposed method consistently achieves superior performance compared to a range of model-free and model-based baselines, demonstrating impressive sample efficiency.
4.  The paper includes a valuable ablation study that analyzes the contribution of the core components.

**Weaknesses:**

1.  The primary contribution appears to be an adaptation of the DreamerV3 to the multi-agent domain by applying the Centralized Training with Decentralized Execution (CTDE) paradigm. The core components—a recurrent model, a representation model, and decoders—are standard in DreamerV1-3.
2.  A central claim is that DMAWM learns "disentangled" latent states. However, the evidence provided is limited to a single visualization in Figure 4. This claim would be much stronger with more rigorous support, such as more diverse qualitative examples showing how the model handles specific agent interactions.
3.  The related work section misses two recent model-based MARL algorithms[a-b]. A direct comparison would be crucial for accurately positioning DMAWM's contribution within the most relevant literature.

a. Liu, Qihan, Jianing Ye, Xiaoteng Ma, Jun Yang, Bin Liang, and Chongjie Zhang. "Efficient multi-agent reinforcement learning by planning." arXiv preprint arXiv:2405.11778 (2024).
b. Ma, Chengdong, Aming Li, Yali Du, Hao Dong, and Yaodong Yang. "Efficient and scalable reinforcement learning for large-scale network control." Nature Machine Intelligence 6, no. 9 (2024): 1006-1020.

**Questions:**

1.  Could you please clarify the contribution of DMAWM beyond being a multi-agent adaptation of the Dreamer architecture?
2.  The claim of learning disentangled latent states is a cornerstone of your work. Could you provide more compelling evidence to support this? For example, could you show latent space interpolations or provide visualizations from SMAC or SMACv2?
3.  Why were a and b not included as a baseline? A comparison would seem essential to evaluate the state of the art.

a. Liu, Qihan, Jianing Ye, Xiaoteng Ma, Jun Yang, Bin Liang, and Chongjie Zhang. "Efficient multi-agent reinforcement learning by planning." arXiv preprint arXiv:2405.11778 (2024).
b. Ma, Chengdong, Aming Li, Yali Du, Hao Dong, and Yaodong Yang. "Efficient and scalable reinforcement learning for large-scale network control." Nature Machine Intelligence 6, no. 9 (2024): 1006-1020.

---

> ### Author Response · Authors · 2025-11-21
>
> Thanks for your valuable feedback! We address your questions and concerns below.
>
> > **Q1: The contribution of DMAWM beyond being a multi-agent adaptation of the Dreamer architecture**
>
> **A1:** Thank you for the question. While DMAWM adopts the Dreamer backbone, it addresses the unique challenges of multi-agent reinforcement learning. Specifically, DMAWM enables decentralized execution without sacrificing the ability to model complex agent interactions.
> Existing model-based MARL methods often compromise on this: they either rely on communication during execution to maintain consistent latent states (e.g., MAMBA) or require reconstructing observations for decision-making (e.g., MBVD).
>
> Our key contributions beyond Dreamer include:
> 1.  Decoupled architecture: We design a novel architecture that consists of a shared environment module and independent agent modules.
> During imagination, all the agent modules simultaneously interact with the environment module. The environment module not only captures the agent-environment interaction but also agent-agent interaction, generating coherent multi-agent imaginary rollouts for policy training.
> 2.  Disentanglement objective: We propose a training objective that aligns the environment module's prior distribution over joint latent states with the factorized posterior of the agent modules. This ensures that the learned latent representations remain disentangled and suitable for decentralized control.
>
> > **Q2: Evidence of disentanglement.**
>
> **A2:** Thank you for the insightful question. We designed an experiment to provide direct evidence that the learned latent states are disentangled across agents.
> Disentanglement is crucial for fully decentralized execution. When latent states are entangled, an agent's state can be unrealistically influenced by others through spurious correlations. For example, even if one agent is unobservable to another, its actions might still propagate through the entangled world model and incorrectly affect the other agent's latent state. Consequently, agents trained in an entangled world model may fail during decentralized execution, as these dependencies cannot be maintained.
> To probe whether this information leakage occurs, we compare DMAWM with MAMBA. For each algorithm, we first train the world model on data generated by random policies for 400K steps on the protoss_5_vs_5 map of SMACv2. We then freeze the world model and train an action predictor network $p\_{\theta}(a^{-i}_t \mid I^i\_{t}, \hat{I}^i\_{t+1})$ to predict the actions of other agents $a^{-i}_t$ at timestep $t$, given the $i$-th agent's current latent state $I^i_t = (h^i_t, z^i_t)$ and next latent state $\hat{I}^i\_{t+1} = (\hat{h}^i\_{t+1}, \hat{z}^i\_{t+1})$ generated by the world model. As a baseline, we also train an observation-based predictor $p\_{\theta}(a^{-i}_t \mid o^i_t, o^i\_{t+1})$ using the agent's current and next observations. The action predictors are trained for 200K steps.
> If the latent states are disentangled, an agent's latent state should not contain more information about other agents than what is inferable from its local observations. Therefore, we expect the latent-based prediction accuracy to be similar to the observation-based baseline. Conversely, if information leakage (entanglement) exists, the latent-based predictor should significantly outperform the baseline by exploiting global information implicitly encoded in the latent state.
>
> Our results in the table below confirm this hypothesis. For DMAWM, the latent-based predictor accuracy (42.3%) is comparable to the observation-based baseline (40.0%). In contrast, for MAMBA, the latent-based predictor achieves significantly higher accuracy (85.0%), indicating severe information leakage. This confirms that DMAWM effectively maintains disentangled latent representations suitable for decentralized execution.
>
> | Method | Action Predictor | Action Prediction Accuracy |
> | :--- | :--- | :---: |
> | Observation | $p\_{\theta}(a^{-i}_t\mid o^i_t, o^i\_{t+1})$ | 40.0\% |
> | DMAWM | $p\_{\theta}(a^{-i}_t\mid I^i_t, \hat{I}^i\_{t+1})$ | 42.3\% |
> | MAMBA | $p\_{\theta}(a^{-i}_t\mid I^i_t, \hat{I}^i\_{t+1})$ | 85.0\% |
>
> > **Q3: Missing baselines (Liu et al., Ma et al.).**
>
> **A3:** Thank you for these references.
> Although both are model-based MARL methods, Liu et al. operates under the CTCE paradigm, while DMAWM is designed for the standard CTDE setting. We have already included MAMBA as a baseline for this category. Ma et al. focuses on scalable network control, which represents a different domain and it is not directly applicable to the more general Dec-POMDP setting.
> We acknowledge the contributions of these works and will add a discussion of them to the related work section.

---

> > ### Comment · Reviewer_JXMY · 2025-11-21
> >
> > Thanks for the authors’ response; it partially addresses my concerns. Could the authors provide some visualization results? Also, could they discuss under what circumstances decoupling is necessary? If we fully decouple, how is this different from using a single agent?

---

> > > ### Author Response · Authors · 2025-11-22
> > >
> > > Thank you for the follow-up questions. Before proceeding with the visualization, we would like to first clarify the question on decoupling to better address your concerns.
> > >
> > > > **Q4: Could you discuss under what circumstances decoupling is necessary?**
> > >
> > > **A4:** Decoupling (not full decoupling as will be explained later) is crucial for decentralized execution. When interacting with the real environment, an agent updates its latent state based solely on its local observations, without direct access to teammates' latent states or actions.
> > > In contrast, during imagination in a centralized multi-agent world model, the model predicts the next joint latent states $\hat{I}^{1:n}_t$ using the previous joint latent states $\hat{I}^{1:n}\_{t-1}$ and joint actions $a^{1:n}\_{t-1}$. If the centralized world model is not trained properly, the latent states of agents can be entangled without respecting the decentralized nature of agents. If policies are trained in such a tightly coupled world model, they would rely on such entangled latent states. When they are deployed in a decentralized manner, they would not have access to such entangled latent states, leading to failure.
> > > However, it is important to note that the latent states of agents are not fully decoupled during execution. When one agent is observable to another, its actions affect the environment and subsequently the other agent's observations, thereby influencing the other agent's latent state. When an agent's actions do not have effects on other agent's observations, their latent states are independent at the next timestep.
> > > Therefore, DMAWM aims to learn a loosely coupled world model, rather than a fully decoupled world model. It learns to dynamically determine when to couple latent states and when to keep them independent, guided by the loss function in Equation 4. Figure 4 demonstrates that latent states become coupled when an agent's action is observable to others, allowing us to decode its actions (e.g., moving and beaming) from the latent state of the observing agents. Conversely, as discussed in our response to Q2, we provide evidence that latent states remain decoupled when agents are not mutually observable.
> > >
> > > > **Q5: If we fully decouple, how is this different from using a single agent?**
> > >
> > > **A5:** Yes, a fully decoupled world model is equivalent to a single-agent world model, as it updates an agent's latent state based solely on its own latent states and actions.
> > > By treating other agents as part of the environment, a fully decoupled world model cannot generate coherent imaginary multi-agent rollouts to model their interactions. This is crucial for tasks that require agent coordination.
> > > In Section 5.3, we present an ablation study, DMAWM-IP, which replaces the transformer-based interaction predictor with an MLP, thereby fully decoupling the agents' latent states. DMAWM-IP learns significantly slower than DMAWM on SMACv2, suggesting that a loosely coupled world model is indeed necessary for learning coordinated policies.
> > >
> > > ---
> > > We would like to thank you again for raising important questions about our submission. Please feel free to let us know if there are additional concerns or questions.

---

### Official Review · Reviewer_zG7f · 2025-10-28

**Soundness:** 1
**Presentation:** 2
**Contribution:** 2
**Rating:** 2
**Confidence:** 4

**Summary:**

The paper proposes a model-based MARL algorithm. The proposed mutli-agent dynamics model consists of individual agent modules and one environment module to integrate all each agent’s information. Agent policies training is speeded up by using the imaginary trajectories from the learned dynamics model.

**Strengths:**

- The paper is easy to follow;

**Weaknesses:**

- The experiment results only show the performance in 50k steps. How about the longer training results? Can the proposed method achieve the performance as good as the baselines? Currently, the comparison is very suspicious.
- How accurate is the proposed dynamics model? It should be quantified.
- Since you are still using centralized training and decentralized execution, why not learn a centralized dynamics model?

**Questions:**

See the weaknesses.

---

> ### Author Response · Authors · 2025-11-21
>
> Thanks for your valuable feedback! We address your questions and concerns below.
>
> > **W1: How about the longer training results than 50K steps?**
>
> **A1:** We kindly clarify that all algorithms were trained for 400K environment steps, not 50K, for all maps in experiments. 400K steps is a reasonable training budget for model-based MARL, which is comparable to the number reported by MBVD and MAMBA. We didn't tune the training steps for each map to ensure a fair comparison.
> Within this budget, DMAWM significantly outperforms all baselines on SMAC. Notably, on the more challenging SMACv2 and Melting Pot benchmarks, DMAWM achieves performance comparable to or better than MAPPO, despite MAPPO being trained for $10\times$ more data ($4 \times 10^6$ steps). This highlights the superior sample efficiency of our method.
>
> > **W2: Accuracy of the dynamics model.**
>
> **A2:** Thank you for the question. While our original submission demonstrated the model's accuracy qualitatively through long-horizon imagination visualizations (Figure 4), we agree that a quantitative evaluation makes a good complement.
> In the table below, we provide an evaluation of the dynamics model by measuring the cumulative observation reconstruction error over varying imaginary rollout lengths on the Coop Mining map. We use the mean squared error (MSE) to measure the reconstruction error.
> The results show that the error remains low for the first 16 steps. The increase of error at longer horizons is expected due to the environment's stochasticity (randomized ore positions), which aligns with the visualizations in Figure 4.
>
> | Rollout steps | 4 | 8 | 12 | 16 | 20 | 24 | 28 | 32 | 36 | 40 |
> | :--- | :---: | :---: | :---: | :---: | :---: | :---: | :---: | :---: | :---: | :---: |
> | Cumulative reconstruction error | 0.0215 | 0.0392 | 0.0752 | 0.1160 | 0.1820 | 0.2599 | 0.3585 | 0.4560 | 0.5784 | 0.6943 |
> | Difference | 0.0215 | 0.0177 | 0.0360 | 0.0408 | 0.0660 | 0.0779 | 0.0986 | 0.0975 | 0.1224 | 0.1159 |
>
> > **W3: Why not learn a centralized dynamics model?**
>
> **A3:** We do employ a centralized dynamics model during imagination to explicitly capture agent interactions, where the interaction predictor of the environment module models the joint stochastic latent states $\hat{z}^{1:n}_t$ conditioned on the joint deterministic states $h^{1:n}_t$, i.e., $\hat{z}^{1:n}_t\sim p\_\phi(\cdot\mid h^{1:n}_t)$.
> However, the environment module is only utilized during centralized training. For execution, the agent modules update the latent state for each agent independently in a fully decentralized manner.

---

### Official Review · Reviewer_kHuN · 2025-10-31

**Soundness:** 2
**Presentation:** 1
**Contribution:** 2
**Rating:** 2
**Confidence:** 4

**Summary:**

This paper proposes DMAWM, a world-model-based multi-agent RL framework that factorizes dynamics into agent modules and a shared environment module. The model learns latent representations and trains decentralized policies through imagined rollouts. Experiments span SMAC, SMACv2, and Melting Pot, where the method reports improved sample efficiency over several baselines.

**Strengths:**

1. Addressing latent imagination for multi-agent control is timely and important, and indeed world models provide compelling benefits for sample efficiency.

2. The paper evaluates on diverse benchmarks, including visual observations, which strengthens empirical relevance.

3. Visualization results are informative and helpful.

**Weaknesses:**

#### **1. (Major) Implicit reliance on CTDE not acknowledged or contextualized**

The method implicitly relies on **centralized training** with latent state alignment, transformer-based global critics, and joint priors over agent latent, which is a classical **CTDE** setup. Yet, the paper **never explicitly acknowledges** this design choice or position itself within the broad literature of CTDE, nor discusses limitations imposed by CTDE on scalability.

#### **2. (Major) Missing comparisons to closely related SOTA baselines**

The paper does **not** compare against nor cite several closely related and highly relevant approaches:
- **LIMARL** (uses agent modules for latent inference and an environment module for state representation, and it is a decentralized execution approach)
- **MA2E** (uses transformers for trajectories learning, and it is a decentralized execution approach)
- **COLA** (latent-space coordination via contrastive objectives, and it is a decentralized execution approach)

These works directly target **latent representations, interaction modeling, and decentralized execution**, overlapping significantly with the stated contribution. This omission is significant and weakens the contribution claim. Thus the related work discussion and experimental baselines are incomplete.

#### **3. Scalability concerns not addressed**

Transformer-based interaction prediction over per-agent embeddings scales **quadratically** in the number of agents. No discussion or experiments address this problem.

#### **4. Overfitting of latent dynamics acknowledged but unaddressed**

The paper admits the latent dynamics "overfit to trajectories generated by the learned policies" (Conclusion), but provides no mitigation strategies. This is a fundamental weakness for world models.

#### **5. Lack of theoretical analysis**

The paper provides no theoretical analysis regarding the proposed disentanglement objective, latent factorization, or convergence properties of imagination-based policy learning.

#### **6. High variance raises concerns about statistical significance**

Some reported results show high variance across training runs (e.g., Figure 2, corridor), making it difficult to assess whether the observed improvements are statistically meaningful rather than due to stochasticity. Given the sample-efficiency claims, more rigorous reporting, such as confidence intervals or statistical tests, would be necessary to show significance.

#### **Reference**

Kharrat, Salma, et al. "Latent Inference for Effective Multi-Agent Reinforcement Learning under Partial Observability." EWRL 2025.

Kang, Sehyeok, et al. "MA $^ 2$ E: Addressing Partial Observability in Multi-Agent Reinforcement Learning with Masked Auto-Encoder." ICLR 2025.

Xu, Zhiwei, et al. "Consensus learning for cooperative multi-agent reinforcement learning." AAAI 2023.

**Questions:**

1. Can the authors clearly state the extent to which DMAWM relies on centralized training of latent states and the critic, and how this differs in practice from standard CTDE architectures?

2. Why are COLA, MA²E, LIMARL, and others omitted from discussion and comparison, given that all use latent representations and support decentralized execution?

3. LIMARL explicitly factorizes dynamics into a local latent representation module and a global environment module. Can the authors comment on the similarity between LIMARL and DMAWM, and clarify the conceptual and architectural differences?

4. How does the transformer-based interaction scale as the number of agents increases, both in computational cost and memory? What are the practical limits of this design?

5. Some reported results (e.g., corridor in Figure 2) exhibit very high variance. Can the authors explain why such large performance fluctuations occur?

6. If the latent dynamics overfit to policy-generated trajectories, as acknowledged, what prevents compounding model errors during imagination from degrading policy learning quality over long horizons?

---

> ### Author Response · Authors · 2025-11-21
>
> Thanks for your valuable feedback! We address your questions and concerns below.
>
> > **Q1: Implicit reliance on CTDE.**
>
> **A1:** We acknowledge that DMAWM operates under the centralized training and decentralized execution (CTDE) paradigm and will explicitly clarify this in an updated version of the manuscript. Specifically, during training, both the shared environment module and the centralized critic utilize the joint deterministic latent states of agents for the stochastic latent state prediction and value estimation, respectively. In contrast, during execution, agents update their latent states and select actions in a fully decentralized manner.
>
> > **Q2: Missing related works (LIMARL, MA2E, COLA).**
>
> **A2:** Thank you for suggesting these works. They are indeed related to DMAWM in that they all leverage latent representations and support decentralized execution, but they belong to a different category.
> LIMARL, MA2E, and COLA are model-free methods that focus on learning more informative latent representations for decentralized execution: COLA aligns an agent’s latent representation with those of its teammates, LIMARL aligns latent representations with a shared latent global state, and MA2E reconstructs teammates’ trajectories based on the latent representations of remaining agents.
> In contrast, DMAWM is a model-based approach that learns a generative dynamics model of the multi-agent environment. The agents’ latent representations are trained to be predictive of their local observations, and the policies are fully trained on the imaginary rollouts generated by the learned generative dynamics model.
>
> > **Q3: Similarity and difference between DMAWM and LIMARL**
>
> **A3:** Although both methods utilize the CTDE paradigm and employ architectures with separate individual and global networks, they differ fundamentally in their modeling objectives and latent variable usage.
> DMAWM is a model-based approach that learns a generative dynamics model of the multi-agent environment. It factorizes the environment state into individual agent latent states. Each agent's latent state is trained to be predictive of its local observations, while the joint latent states are used to predict the environment state at the next timestep. This learned dynamics model generates imaginary rollouts for policy learning.
> In contrast, LIMARL is a model-free method that does not model environment dynamics. Its global network, the State Representation Module (SRM), uses a VAE to encode the current ground-truth global state into a latent representation. The local network, the Recurrent Observation-to-Latent Inference Module (ROLIM), infers a latent representation from each agent's local history to align with the SRM's encoding. This inferred latent variable is then used to augment the individual value function for action selection.
>
> > **Q4: Scalability of the transformer-based interaction.**
>
> **A4:** We acknowledge that the transformer's computational complexity scales quadratically with the number of agents. However, this cost remains manageable within the scope of standard benchmarks like SMAC, where the number of agents ranges from 2 to 27 (our experiments use a maximum of 25). For significantly larger systems involving hundreds of agents, efficiency could be improved by adopting sparse or linear attention mechanisms, though exploring these optimizations is beyond the scope of this work.
>
> > **Q5: High variance on the Corridor map.**
>
> **A5:** The Corridor map presents a significant exploration challenge, leading to variability in how quickly agents solve the task across different seeds. The observed high variance stems from differences in the time required for convergence rather than fluctuation due to training instability. Specifically, at 400K steps, two seeds had not yet solved the task (success rate < 10%). However, by 500K steps (we continued training for 100K steps), all seeds achieved a success rate above 60%, resulting in a mean win rate of 76.2% with a significantly reduced standard deviation of 12.7%.
>
> > **Q6: Overfitting of latent dynamics.**
>
> **A6:** As noted in our future work section, world models can suffer from overfitting to the specific data distribution induced by the sampling policy, potentially limiting generalization to unseen policies. This typically occurs because policy entropy decreases during convergence, leading to a narrower distribution of collected data. However, this is acceptable in our setting because the sampling policy is identical to the policy being trained. To further improve robustness in future work, one could train a population of diverse policies to broaden the data distribution for world model training.

---

> > ### Comment · Reviewer_kHuN · 2025-11-27
> >
> > *Thank you for the clarifications.* While the rebuttal acknowledges several points and partially addresses some concerns, the major issues remain largely unresolved.
> >
> > **CTDE reliance**: Acknowledging the CTDE setting is appreciated, but the method’s contribution still appears overstated relative to existing CTDE architectures. The current manuscript does not clearly position DMAWM within this literature or discuss CTDE-related limitations. *I assume the authors will make an effort to clarify this in a revised draft.*
> >
> > **Missing related work:** The distinction between model-based and model-free methods does not justify omitting LIMARL, MA²E, and COLA from discussion or comparison. These approaches share core aims, including latent representations, factorization, and decentralized execution, and address the same environments. Even if methodologies differ, algorithms solving the same setting warrant citation and empirical comparison.
> >
> > **Similarity to LIMARL:** The rebuttal’s differentiation does not fully address the conceptual overlap. LIMARL **does** model environment dynamics through latent-state prediction from observations, and it **also uses local and environment modules**. The key difference lies in how the predicted latent states are used (as network inputs vs. imaginary rollouts), **not in whether dynamics are modeled.** An empirical comparison would be necessary to substantiate claims of novelty or superiority.
> >
> > **Scalability:** The authors note the quadratic cost of the transformer and state it is manageable for SMAC-sized problems. However, **SMAC is a benchmark, should not be the end goal**.
> >
> > **High variance:** The explanation (exploration difficulty) does not resolve concerns about stability or statistical significance. Related methods such as LIMARL and MA²E solve the Corridor environment with higher win rates and substantially lower variance, suggesting this issue may be method-specific.
> >
> > **Theory** (minor): The rebuttal does not acknowledge or address the absence of theoretical justification for the latent factorization or the training objectives.
> >
> > ***Although the rebuttal offers helpful clarification, major issues remain largely unresolved, and my rating therefore remains unchanged.***

---

### Official Review · Reviewer_QkZz · 2025-10-31

**Soundness:** 3
**Presentation:** 2
**Contribution:** 3
**Rating:** 6
**Confidence:** 2

**Summary:**

The work proposes Disentangled Multi-Agent World Model (DMAWM), a framework to learn decentralized policies using model-based techniques. The proposed architecture learns decentralized policies in the latent space, featuring independent agent modules and a shared environment module that jointly learn a factorized latent representation that captures interactions between the agents while disentangling individual agent latent spaces. The authors perform experiments over different scenarios/benchmarks, comparing the performance of the proposed method against other baselines.

**Strengths:**

The paper features a good discussion of related work in Sec. 2. The experimental protocol seems solid, with the authors performing multiple runs and reporting the standard deviation of the results. The experimental results seem reproducible. The experiments are also extensive, considering several different environments. The baselines selected in the empirical part of the work seem representative and diverse. The experimental results seem to be in favor of the proposed method. The authors also provide an ablation study. The work seems to feature a sufficient degree of novelty for publication, even though I am not very familiar with previous works.

**Weaknesses:**

I feel like the clarity of the paper could slightly be improved, particularly in Sec. 4. I understand that the method proposed (DMAWM) features many components and that the authors seem to have tried to come up with a graphical way to explain how these different components interact (Fig. 1). Still, it took me a while to understand how all the pieces come together (and I read the section multiple times).

**Questions:**

- in Fig. 1 (b), why did you use $\hat{I}$ instead of $I$?
- "Since no observation is available during imagination, the interaction predictor takes the joint deterministic state as input to model the interaction between agents instead." - could the authors clarify this sentence? Do you assume access to the underlying state during imagination? Shouldn't the imagination part be entirely done in a recursive manner by iteratively predicting latent states?
- line 288, sentence " After that, the interaction predictor samples the (...)". As far as I understood, while the policies of each agent are decentralized in the sense that they only rely on local information for each of the agents, this interaction predictor runs in a centralized fashion, right? This is because it depends on the h's of all agents and, therefore, either the agents all have access to the interaction predictor and communicate with each other in order to sample the joint stochastic state, or there exists a central node sampling the joint stochastic state and broadcasting it to all agents. Is my reasoning correct?
- Sec. 5: So, in practice, the policies of DMAWM are trained every 32 env. steps using 1024 parallel rollouts generated from the model. Do you also use this exact same rate for the other model-based baselines? If so, I think it is important to emphasize that this is the case in the document so that we can fairly compare the different model-based approaches.

---

> ### Author Response · Authors · 2025-11-21
>
> We thank the reviewer for the positive assessment and the constructive feedback. We address your questions and concerns below.
>
> > **Q1: In Fig. 1 (b), why did you use $\hat{I}$ instead of $I$?**
>
> **A1:** We apologize for the confusion. We use the notation $\hat{I}$ to explicitly denote imaginary latent states, distinguishing it from $I$, which represents information derived from real observations. In Figure 1(b), the model is rolling out in the latent space (imagination), so the states and interactions are predicted by the model rather than observed from the environment. We will clarify this notation in an updated version of the manuscript.
>
> > **Q2: Could the authors clarify this sentence? Do you assume access to the underlying state during imagination? Shouldn't the imagination part be entirely done in a recursive manner by iteratively predicting latent states?**
>
> **A2:** We do not assume access to the underlying ground-truth state during imagination. The interaction predictor takes the joint deterministic state $h^{1:n}_t$ as input to predict the joint stochastic state $\hat{z}^{1:n}_t$. The imagination process is indeed fully recursive: Given the joint latent states $(\hat{h}^{1:n}\_{t-1}, \hat{z}^{1:n}\_{t-1})$, the agent modules update the latent states for each agent $\hat{h}^i_t = f\_\phi(\hat{h}^i\_{t-1}, \hat{z}^i\_{t-1}, a^i\_{t-1})$, and the interaction predictor generates the joint stochastic states for all agents $\hat{z}^{1:n}_t\sim p\_\phi(\cdot\mid \hat{h}^{1:n}_t)$.
>
> > **Q3: Either the agents all have access to the interaction predictor and communicate with each other in order to sample the joint stochastic state, or there exists a central node sampling the joint stochastic state and broadcasting it to all agents. Is my reasoning correct?**
>
> **A3:** Yes, your understanding is correct for the centralized training phase. During training (specifically, during imagination), the interaction predictor functions as a central component that samples the joint stochastic states for all agents to model their interactions, i.e., $z^{1:n}_t \sim p\_\phi(\cdot\mid h^{1:n}_t)$.
> However, it is important to note that this centralized access is restricted to training. During decentralized execution, agents do not communicate or access the interaction predictor. Instead, they update their latent states independently using their agent modules, i.e., $z^i_t \sim p\_\psi(\cdot\mid h^i_t, o^i_t)$.
>
> > **Q4: Do you also use this exact same rate for the other model-based baselines?**
>
> **A4:** We aimed to align the training rate on imaginary rollouts across baselines to ensure a fair comparison, although implementation details vary by algorithm.
> For MAMBA, training occurs at the end of each episode. It performs $4$ epochs of imaginary training, with each epoch generating $720$ parallel imaginary rollouts. Since most SMAC and SMACv2 maps evaluated in our experiments end in fewer than $70$ steps, MAMBA effectively has a slightly higher training rate than DMAWM on these maps.
> In contrast, MBVD does not fully train the agent on imaginary rollouts; instead, it uses them as auxiliary inputs for the mixing network, resulting in a lower training rate on imaginary rollouts.

---

> > ### Comment · Reviewer_QkZz · 2025-11-23
> >
> > I thank the authors for their response, which addressed and clarified my concerns.

---

### Note · Authors · 2025-12-04

I have read and agree with the venue's withdrawal policy on behalf of myself and my co-authors.